# Genome-Wide Identification and Expression Analysis of the SUT Family from Three Species of Sapindaceae Revealed Their Role in the Accumulation of Sugars in Fruits

**DOI:** 10.3390/plants13010095

**Published:** 2023-12-28

**Authors:** Sirong Jiang, Pengliang An, Chengcai Xia, Wanfeng Ma, Long Zhao, Tiyun Liang, Qi Liu, Rui Xu, Dongyi Huang, Zhiqiang Xia, Meiling Zou

**Affiliations:** 1College of Tropical Crops, Hainan University, Haikou 570228, China; 20090100110019@hainanu.edu.cn (S.J.); 21220951310001@hainanu.edu.cn (P.A.); 23110901000043@hainanu.edu.cn (C.X.); 21220951310058@hainanu.edu.cn (W.M.); longz0322@hainanu.edu.cn (L.Z.); 22220951310169@hainanu.edu.cn (T.L.); 22110901000041@hainanu.edu.cn (Q.L.); 23210901000063@hainanu.edu.cn (R.X.); 2Hainan Yazhou Bay Seed Laboratory, Sanya Nanfan Research Institute, Hainan University, Sanya 572025, China

**Keywords:** sucrose transporters, *Sapindaceae Juss*, *SUT* gene family, *SUT* gene expression

## Abstract

*Sapindaceae* is an economically important family of Sapindales and includes many fruit crops. The dominant transport and storage form of photoassimilates in higher plants is sucrose. Sucrose transporter proteins play an irreplaceable role in the loading, transportation, unloading, and distribution of sucrose. A few *SUT* (sugar transporter) family genes have been identified and characterized in various plant species. In this study, 15, 15, and 10 genes were identified in litchi, longan, and rambutan, respectively, via genome-wide screening. These genes were divided into four subgroups based on phylogenetics. Gene duplication analysis suggested these genes underwent potent purifying selection and tandem duplications during evolution. The expression levels of *SlSut01* and *SlSut08* were significantly increased in the fruits of *Sapindaceae* members. The homologs of these two genes in longan and rambutan were also highly expressed in the fruits. The expression pattern of *SUT*s in three organs of the two varieties was also explored. Subcellular colocalization experiments revealed that the proteins encoded by both genes were present in the plasma membrane. This report provides data for the functional study of *SUT*s in litchi and provides a basis for screening sugar accumulation-related genes in fruits of *Sapindaceae*.

## 1. Introduction

Sugar, which determines the quality and flavor of fruits, is the primary nutrient for humans. Soluble sugars in fruits mainly include fructose, sucrose, and glucose [1]. Fruits can be divided into hexose, sucrose, or intermediate accumulation types considering the sugar composition of the mature fruits [2,3]. Fructose metabolism is a complex process [4]. Sugar metabolism can be classified based on the substrates sorbitol, sucrose, hexose, and starch [5]. Sucrose is crucial for plant growth and development and, as a central metabolite, has other related functions, such as a signaling molecule [6,7].

Sugar transporters (*SUT*s) and sugar transport proteins (*STP*s) act as symporters, while “Sugars Will Eventually Be Exported Transporters” (*SWEET*s) act as uniporters [8]. SUTs exist in tissues and cells of higher plants and are important participants in the transmembrane transport and distribution of apoplasts [9,10]. SUTs belong to the major facilitator superfamily of transmembrane proteins and are phylogenetically divided into types I, II, and III [11]. *SUT*s are encoded by a gene family, which usually consists of three to nine members [12]. Three *SUT*s, namely, *SlSut1* (*solyc11g017010.1.1*), *2* (*solyc05g007190.2.1*), and *4* (*solyc04g076960.2.1*), have been identified in tomato [13]. *SUT*s have also been identified in other species. A total of 49 sugar transporter gene sequences belonging to *MST* (45) and *SUC*/*SUT* (4) were identified in the pineapple genome [14]. A total of nine *SUT*s were identified in the beet genome and divided into groups 1, 2, and 3, which were unevenly distributed on four chromosomes [15]. In total, 12 SUTs were discovered in D. farinosus using whole-genome identification [16]. Using the hidden Markov model (HMM) profile (PF13347.6) and BLASTp, 14 *SUT*s were identified in *C. annuum* Cvs zunla and CM334, *S. lycopersicum*, *S. melongena*, and *S. tuberosum* [17].

*Sapindaceae* is distributed in tropical and subtropical regions, of which several are common fruit plants, such as litchi (*Litchi chinensis*), longan (*Dimocarpus longan*), and rambutan (*Nephelium lappaceum*). Several studies on gene families involved in the sucrose synthesis pathway in litchi have been conducted. For example, 16 *LcSWEET*s were identified and named based on their homologs in *Arabidopsis thaliana* and *Vitis vinifera* [18]. Four *SPS*s—*LcSPS1*, *2*, *3*, and *4*—were isolated from litchi [19]. The genomic organization analysis indicated the four litchi *SPS* genes have very similar exon–intron structures. However, there is no information about the *SUT* family genes in litchi and other species of Sapindaceae. We know nothing about the function and structure of the *SUT* family genes in Sapindaceae. Therefore, this study was conducted to identify the *SUT*s, ascertain their crucial functions, analyze their evolution, and provide a theoretical basis for sugar accumulation in the fruits of *Sapindaceae*. The expression pattern for all *SUT*s and the key genes regulating the sugar content was ascertained by accessing the RNA-seq databases of litchi fruits. This study may reveal the involvement of these *SUT*s in fruit development and sugar quality conformation in litchi, provide genetic resources for their genetic improvement in the future, and act as a reference for the regulation and improvement in the quality and quantity of sugars in other related fruit-bearing species.

## 2. Results

### 2.1. Phylogeny and Classification of SUTs

In total, 15, 10, 15, and 8 *SUT*s were identified from the whole genomes of litchi, longan, rambutan, and grape, respectively, using a combination of *SUT* family gene sets obtained by HMM prediction and BLAST-based comparison. The *SUT*s in litchi, longan, and rambutan were termed *SlSut*, *DlSut*, and *NlSut*, respectively (Appendix A). A phylogenetic tree was constructed based on all the amino acid sequences identified in these five species to classify them and establish their evolutionary relationship. Since the classification of *SUT*s from *Arabidopsis thaliana* was unambiguous, the evolutionary tree was divided into four branches, named I, II, III, IV-1, and IV-2, based on the ID of *A. thaliana* (Figure 1).

### 2.2. Chromosomal Localization of SUTs

The positions of *SUT*s on the different chromosomes of litchi, longan, and rambutan were determined to determine the distribution of *SUT*s. The ten *SUT*s in longan were mapped to chromosomes 6, 9, 10, 12, 14, and 16 (Figure 2B; Appendix A). Of the 15 *SUT*s in rambutan, 2 were not mapped to the chromosome, and the rest were mapped to the corresponding chromosome (Figure 2C; Appendix A). Litchi and rambutan had the same number of *SUT* family genes, which were located on 10 chromosomes (Figure 2A; Appendix A). In longan, the genes were located on fewer chromosomes and distributed unevenly, and most genes were distributed on Chr9 (4 genes). There were 13 genes in rambutan distributed on 7 chromosomes, only 1 gene on Chr1, Chr5, Chr13, and Chr14, and Chr6 had the most chromosome genes (5). The *SUT* family genes of litchi were evenly distributed on its 10 chromosomes. *SlSut05*, *SlSut06*, *SlSut07*, and *SlSut08* were distributed on Chr07, with two genes on SL06 and SL14, and one gene on the other seven chromosomes.

### 2.3. Structure, Conserved Motifs, and Domain Analysis of SUTs

The analysis of gene structure, conserved motifs, and domain distribution revealed the conserved pattern of *SUT* family genes. Similar gene structures, conserved motifs, and domains were detected in the same group or subgroup. All 40 *SUT* family genes from 3 species were compared in structure and motif (Figure 3A,B). Almost all the *SUT*s had at least two exons, except a few, such as *NlSut*03 in group I, which had only one exon. Motif distribution was similar in each group. For example, the motif composition in group I was the same, the motif distribution of all genes in group III was also similar, and the motif distribution in group IV was looser. The functional prediction and analysis of the domains of the genes from the three *Sapindaceae* species indicated that most belonged to the GPH_sucrose, followed by the MFS, and few to the MFS_MFSD5_like superfamilies, among which MFS_SLC46_TetA_like, MFS_HMIT_like, and MFS_GLUT6_8_Class3_like occurred only in the *SUT*s of litchi (Figure 3C; Appendix A). 

The cis-element in the promoter of a gene is crucial for initiating its expression. In total, 14 cis-elements were identified in *SUT*s, divided into 4 types: plant hormone, environmental stress, light response, and metabolic-responsive elements (Appendix A). The promoters of almost all *SUT*s contained at least one environmental stress-responsive element. In total, 67 ABRE abscisic acid-responsive elements were enriched in the *SUT*s, suggesting they may play crucial roles in multiple hormone-related signaling pathways. Additionally, 75 biotic and abiotic stress-responsive elements (AREs) and 39 MBS elements were enriched, suggesting a pivotal role of *SUT*s in the responses to these stresses and in the defense and repair processes associated with trauma and pathogen invasion. In addition, 70 G-box and 17 I-box elements were enriched along with 25 MREs related to the light response, indicating a crucial role of *SUT*s (Figure 3D; Appendix A). The differences in the three light-responsive elements (ACE, MRE, and GATA-MOTIF) in the three *Sapindaceae* species were compared, due to which the cis-elements of ACE could be identified in rambutan and longan but not in litchi. Litchi, rambutan, and longan contained MREs and GATA-MOTIFs. The expression of *SUT*s may be regulated by various cis-elements responsive to light, hormones, defense-related signal transduction, and abiotic stress resistance during the growth of litchi, longan, and rambutan.

### 2.4. Collinearity Analysis of SUTs

A linear relationship comparison was first made to understand the evolutionary relationship of *SUT*s in *Sapindaceae*, revealing 53 homologous gene pairs between litchi and rambutan, 48 between litchi and longan, and 52 between longan and rambutan. In addition, 75 pairs between litchi, 40 pairs between longan, and 49 pairs between rambutan were identified (Figure 4A, Appendix A). The highest number between litchi and longan indicated a close inter-relationship. The pairwise Ka/Ks ratios of each homologous gene pair were 0–1 (163/164) (Figure 4B, Appendix A), indicating a purifying selection. The divergence time of orthologous *SUT*s was estimated (based on their synonymous substitution rates (Figure 4C)) to be 0.44–30 million years for those between litchi and longan, 0.59–30 between litchi and rambutan, and 0.45–28.28 between longan and rambutan (the abnormal value of 185 was excluded).

### 2.5. Evolutionary and Positive Selection Analyses of SUTs

Whole genome duplication (WGD) and segmental duplication events drive the expansion of gene families and determine the functional divergence. Gene duplications can be categorized into singletons, dispersed, proximal, tandem, and WGD/segmental. Of these, dispersed played a critical role in the amplification of *SUT*s in the three species, with seven, six, and six dispersed *SUT*s in litchi, longan, and rambutan, accounting for 70%, 40%, and 40%, respectively. This was followed by tandem, with seven, three, and four genes accounting for 70%, 20%, and 27%, respectively. However, WGD or segmental occurred only in rambutan, proximal only in litchi, and singleton only in longan (Figure 5A, Appendix A).

Using the total SUT protein content to construct a phylogenetic tree, *Arabidopsis thaliana* was identified as the outermost branch, followed by grape, rambutan, litchi, and longan (Figure 5B). Using *SUT*s to reconstruct the tree, grape, and *Arabidopsis thaliana* together were found to form a branch, and rambutan and longan were evolutionarily the closest (Figure 5C). Therefore, the evolution of *SUT*s was not associated with species differentiation. The CAFE tool was used to calculate the expansion and contraction of *SUT*s, which indicated the expansion of *SUT*s in all species, especially rambutan and longan (Figure 5C, Appendix A). During domestication and subsequent improvement, plants were subjected to intensive positive selection for desirable traits. Positive selection analyses are important for understanding gene functions during evolution. Analysis of these *SUT*s using codon substitution models (M0, M1, M7, and M8) of the codeML package suggested that 16 *SUT*s underwent an intense positive selection pressure (Figure 5D, Appendix A).

### 2.6. Analysis of SUTs’ Expression Patterns and Subcellular Localizations

In order to better understand the performance of *SUT* family genes in the three species, we collected RNA-seq data of flowers, fruits, and leaves of litchi, longan, and rambutan, and found that *SUT* family genes were highly expressed in the fruits of the three species (Appendix A). So, to further study the gene expression pattern in the fruits, we chose litchi as an example. Samples of three organs (pulp, peel, and seed) from two varieties were collected for RNA-seq analysis (Appendix A). In the two varieties of litchi, “QingPi” and “MiLi”, most genes were expressed higher in the pulp and peel than in the seed (Figure 6A, Appendix A). Particularly, *SlSut01* and *SlSut08* showed more significant expression. From qRT-PCR (real-time quantitative PCR) experiments for these two varieties, it was found that the expression of *SlSut01* and *SlSut08* genes in the pulp and peel was higher than that in the seed (Figure 6B, Appendix A). Finally, we used tobacco leaves as material, and GFP analysis of *SlSut01* and *SlSut08* were expressed in the cell membrane (Figure 6C, Appendix A). The same results have been found in other species [20]. In addition, regarding the observation of *SlSut01* and *SlSut08* subcellular colocalization, the GFP signal of *SlSut01* and *SlSut08* fusion-expressing cells was clearly visible in the cell membrane. These results confirmed that both *SlSut01* and *SlSut08* were localized in the cell membrane.

## 3. Discussion

With the rapid development of sequencing technology, the genomics-based study of gene families is becoming more popular, and the basis of studying the function of gene family is genomics research. *SUT*s have been reported in many plants [21,22] but not in *Sapindaceae*. Litchi, longan, and rambutan are important tropical fruits [23,24]. Since their genomes have been completely sequenced, an opportunity was provided to study the *SUT*s in these three species of *Sapindaceae*. In this study, 15, 15, and 10 sugar transporter-encoding genes were identified in the seedless litchi, longan, and rambutan, respectively. In addition, nine and eight *SUT*s were identified in *Arabidopsis thaliana* and grapes, respectively, and were compared to those in the three species. In number, these three species of Sapindaceae had obvious expansion compared with grapes and *Arabidopsis thaliana*. The expansion of sugar transporter genes in these three tropical fruits may be an important reason for the increase in sugar content. In this study, three species of Sapindaceae and phylogenetic trees of grape and Arabidopsis were constructed. It can be seen that, after *Arabidopsis thaliana*, Sapindaceae and grapes expanded to form a new *SUT* branch. This *SUT* family gene expansion was probably the key reason for the increase in sugar in fruits. Therefore, the quantitative expansion of the sugar transporter-encoding genes in *Sapindaceae* was closely associated with the high sugar content.

The phylogenetic tree constructed using a single copy of the orthologous genes in the genomes of the five species was closer to the actual evolution of species [25]. The species-based tree of the whole genome sequences indicated a more intimate inter-relationship between litchi and longan. The phylogenetic tree constructed using SUT proteins showed that longan was more closely related to rambutan. Hence, the evolution of *SUT*s in *Sapindaceae* was unrelated to species differentiation. The analysis of the structures, conserved motifs, and domains of these genes revealed that the conserved motifs in the genes of the same group were highly similar and more consistent in positional distribution, with almost identical domain sequences. The exons of the genes of litchi and longan in the same group (such as *DlSut05*, *SlSut06*, etc.) were longer than those of rambutan (*NlSut03*). The results revealed that the exon-containing regions were remarkably expanded after the differentiation of sugar transporter-encoding genes in litchi, longan, and rambutan. This may be one of the reasons behind the genome expansion in litchi and longan. The sizes of the haplotypes of litchi, longan, and rambutan were 515, 438, and 397 Mb, respectively. In addition, they were generally highly conserved in conserved domains, indicating that although there were some differences in the number of sugar transporters in litchi, longan, and rambutan, the gene sequences themselves were relatively conserved in the evolution process. For example, the conserved domain of the GPH_sucrose superfamily was identified in all species. The conserved sequences and structures of sugar transporter-encoding genes in the same subclass may indicate a similarity in their functions.

In this research study, the variations in the number of *SUT*s among litchi, longan, and rambutan were not the only factor for the differences in taste, but the expression of these genes was also one of the crucial components leading to distinct phenotypes [26]. The abundance in the expression of *SUT*s varied in the different organs; for example, *AtSUC3* and *5* had higher expression levels in the mature leaves and seeds during early germination [27]. The *SUT*s were conspicuously expressed in the pulp and peels of litchi, unlike the seeds. qRT-PCR further confirmed these results. Fruit peels must absorb and accumulate sucrose to provide the energy and materials needed to promote fruit development and maturity. Sucrose is a crucial carbohydrate in plants, the primary source of energy for metabolism, and a pivotal compound involved in communication with the external environment. Therefore, an elevated expression level of sucrose transporters in fruit peels helps fruits absorb and accumulate sucrose to maintain the rate of fruit development. The correlation between the RNA-seq and qRT-PCR data confirms the critical roles of *SlSut01* and *08* during sugar accumulation in the fruit.

This study forms a basis for future comprehensive functional validation studies of these genes. The qRT-PCR and subcellular localization analysis [28] indicated that two genes, *SlSut01* and *08*, were present in the cell membrane and played key roles in the accumulation of sugars in litchi fruits.

## 4. Materials and Methods

### 4.1. Identification of SUTs

The genome sequences of litchi, longan [29], and rambutan [30] were accessed. The genome of the seedless litchi as a reference was obtained from a previous study of our group. Since all three species of *Sapindaceae* were fruit-bearing, the genome sequence of grape (*Vitis vinifera*) as the ancient fruit species was employed as a reference [29,30]. As the *SUT*s of *Arabidopsis thaliana* were relatively comprehensive, they were selected as a reference to classify all the genes. In total, nine *SUT*s of *Arabidopsis* have been identified [31]. The protein sequences encoded by these were accessed from Tair (https://www.arabidopsis.org/, accessed on 28 June 2023) and applied for multiple sequence alignments using NCBI BLAST+ (V2.11.0) to identify the *SUT*s in litchi, longan, rambutan, and grape. The e value was <1 × 10^−5^. *SUT*s belong to the MFSs family. Based on the MFS as the retrieval order, the HMM files (PF13347 and PF05631) of the structural domains of *SUT*s were downloaded from Pfam (https://www.ebi.ac.uk/interpro/search/text/ (accessed on 28 June 2023)). Candidate genes from litchi, longan, rambutan, and grape (e value of 1 × 10^−5^) were identified using Hmmsearch (V3.3.2). The members of *SUT*s predicted by the PF number were combined with those obtained by BLAST, and the *SUT*s of several species were obtained. All candidate *SUT* family genes sequences were confirmed using the NCBI conserved domain database (https://www.ncbi.nlm.nih.gov/Structure/cdd/wrpsb.cgi, accessed on 28 June 2023) and the SMART program (http://smart.embl-heidelberg.de/, accessed on 28 June 2023).

### 4.2. Construction of the Phylogenetic Tree 

The protein sequences of *SUT* family genes in five species (litchi, longan, rambutan, grape, and Arabidopsis) were compared by using a muscle tool [32]. The gene tree was constructed with the IQ-TREE (V1.6.12) tool, and the file was visualized on the ITOL website [33,34]. The proteins encoded by the whole genome of five species were compared using OrthoFinder (V2.4.0) software, and the phylogenetic tree was constructed using a single copy of the orthologous genes [35]. Only *SUT*s were used to build the phylogenetic trees, and the differences between the two trees were compared. CAFE (V4.2) software was used to calculate the expansion and contraction of *SUT*s from the five species [36].

### 4.3. Chromosome Localization, Structure, and Conserved Motifs of Genes

The information of the *SUT*s as a GFF file was extracted using TBtools (V2.029) software [37]. The distribution of *SUT*s on each chromosome of rambutan, litchi, and longan was drawn based on the physical location in the GFF file. The structural analysis of *SUT*s in these species was carried out using TBtools by submitting the GFF file and gene ID, thus showing the positions of the exons, introns, and the 5′ and 3′ untranslated regions (UTRs). The amino acid sequences of the proteins encoded by the *SUT*s in litchi, rambutan, and longan were analyzed, and the conserved motifs identified by the multiple expectation maximization (MEME) method were set to default parameters [38].

### 4.4. Analysis of the Sequences of the SUTs

The molecular weights and isoelectric points of the SUT proteins were obtained using the ExPASy server (http://web.expasy.org/compute_pi/, accessed on 28 June 2023), and their subcellular location was determined using Cell-PLoc (http://www.csbio.sjtu.edu.cn/bioinf/Cell-PLoc-2/, accessed on 28 June 2023) software [39]. Multiple sequence alignment was performed using the MUSCLE (V3.8.31) tool, and the domain features of the genes were identified by the WebLogo online tool (http://weblogo.berkeley.edu/logo.cgi, accessed on 28 June 2023) [40]. The cis-acting elements in the promoters were identified through the PlantCARE online server (http://bioinformatics.psb.ugent.be/webtools/plantcare/html/, accessed on 28 June 2023).

### 4.5. Collinearity Analysis and Duplication Events in SUTs

The *SUT*s in litchi, longan, and rambutan were analyzed collinearly by MCScanX (V1.3) software [41]. First, the sequences of their proteins were compared using the BlastP (V2.10.1+) program, with an e-value of 1 × 10^5^. MCScanX was then used to identify the collinear blocks with parameters set to −k: 50, −s: 5, and −m: 25. The type of replication in each gene was classified using the duplicate_gene_classifier program of MCScanX. The information on *SUT*s was extracted from the results file.

### 4.6. Calculation of the Selection Pressure and Analysis of the Evolution of SUTs

WGDI (V0.6.5) software was used to calculate the dissimilarity rate (Ka), synonymy rate (Ks), and ratios (Ka/Ks) of the orthologous gene pairs employing their coding sequences (CDS), thereby estimating the divergence time [42]. WGDI uses C to align the protein sequences. Then, pal2pal.pl converted the protein alignment into the codon alignment based on CDS. Finally, the Ka and Ks were estimated with the YN00 module of PAML (V4.9J). Using a neutral substitution rate of 7.08 × 10^−8^ and the number of substitutions/site/year, the Ks values were employed to estimate the divergence time of the orthologous gene pairs.

The CDSs of the *SUT*s were compared using MUSCLE (V3.8.31). Using CodeML (V4.9J) implemented by PAML, the ratio of the nonsensical to the synonymous distances (ω) was inferred in each branch of the phylogenetic tree [43]. The gap alignment was performed by the method of complete deletion. The gene sequence that produced super large gaps was deleted to reduce the gaps. The differences between the sites were determined using the likelihood ratio test between the M0-M1 and the M7-M8 models.

### 4.7. Analysis of the Expression Patterns of SUTs

To explore the gene expression patterns of the *SUT* family genes, two replicates of Illumina RNA-seq data were generated in three organs of litchi (pulp, peel, and seeds). The RNA-seq data of longan and rambutan were obtained from NCBI. Cutadapt was used to remove the reads containing connectors, polyA, polyG, N > 5%, and of low quality to obtain valid data. The transcripts were compared using HISAT2 (V2.1.0), assembled by StringTie (V1.3.3b) software, and quantified by FPKM [44,45,46].

### 4.8. Subcellular Colocalization of SlSut01 and SlSut08

The subcellular localization of *SlSut01* and *08* was studied by cloning their full-length CDS into the pBWA(V)HS:GFP vector to prepare the *SlSut01*:GFP and *SlSut08*:GFP fusion vectors. These were used to transform the *Agrobacterium* cells, which were injected into the epidermal cells of tobacco leaves and observed under a confocal laser scanning microscope.

### 4.9. RNA Extraction and qRT-PCR 

Litchi samples were collected, frozen in liquid N_2_, and stored in a –80 °C refrigerator for RNA extraction. The MolPure^®^ Plant RNA Kit (Shanghai Yisheng Biotechnology Co. Ltd., Shanghai, China) was used to extract the total RNA. The NovoScript^®^ Plus All-in-one 1st Strand cDNA Synthesis SuperMix (gDNA Purge) (Beijing BioDee Biotechnology Co. Ltd., Beijing, China) was used to synthesize the cDNA for verifying the exon–intron structure. Real-time PCR (Thermo Fisher Scientific, Waltham, MA, USA) was employed to measure the transcription levels of *SUT*s in different samples using primer pairs (Appendix A) designed for each cDNA sample. The program was 95 °C for 10 min, 95 °C for 20 s, 40 cycles, and 60 °C for 60 s. The relative expression levels of *SUT*s were calculated using the ^∆∆^CT method [47], and the statistical significance of the differences was determined by one-way ANOVA.

## Figures and Tables

**Figure 1 plants-13-00095-f001:**
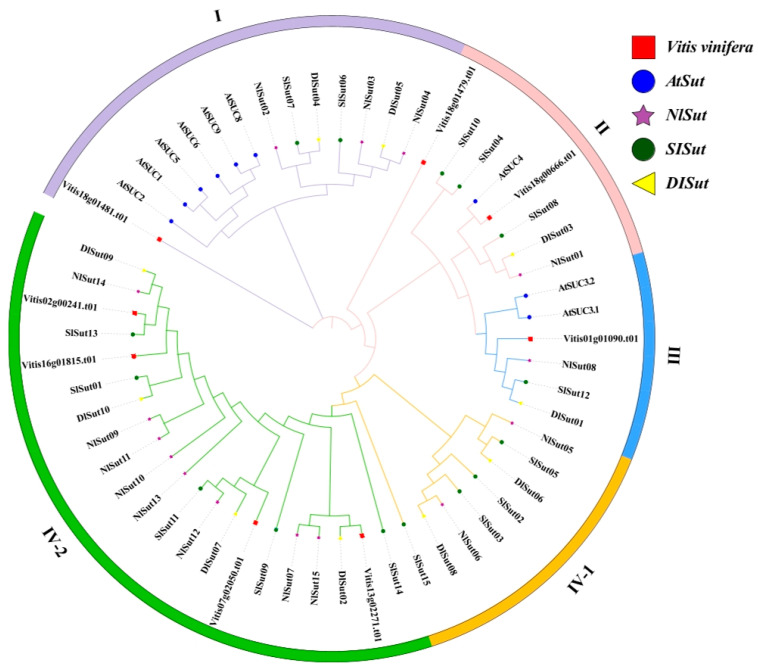
Phylogenetic analysis of *SUT* proteins from five species: *Litchi chinensis* (*SlSut*), *Dimocarpus longan* (*DlSut*), *Nephelium lappaceum* (*NlSut*), *Vitis vinifera* (*Vitis*), and *Arabidopsis thaliana* (*AT*). Phylogenetic analysis of multispecies SUT family proteins was performed, and phylogenetic trees were constructed for 57 *SUT* family genes using the NJ method with 2000 bootstrap replicates. The different shapes represent various species, and the colored circles represent the distinct subclades. All *SUT*s were classified into subfamilies based on that of *AtSut*.

**Figure 2 plants-13-00095-f002:**
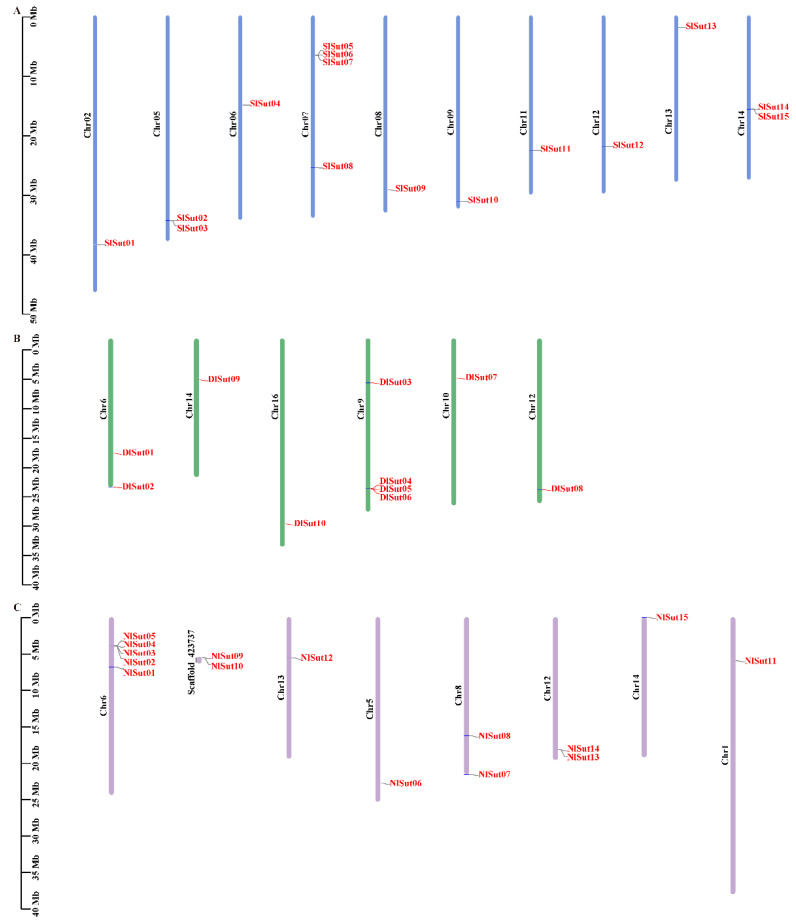
This is a figure. Schemes follow another format. If there are multiple panels, they should be listed as: (**A**) Description of what is contained in the first panel; (**B**) Description of what is contained in the second panel; (**C**) Description of what is contained in the third panel. Figures should be placed in the main text near to the first time they are cited.

**Figure 3 plants-13-00095-f003:**
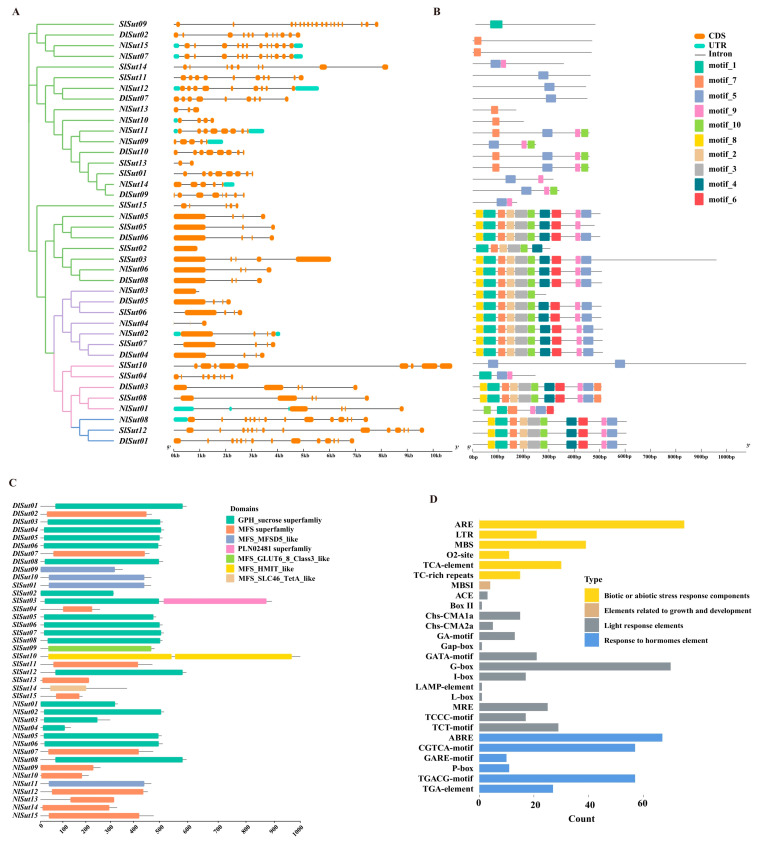
Conserved motifs, structures, and structural domains of *SUT*s. (**A**) Phylogenetic trees of the *SUT*s of litchi, longan, and rambutan were constructed using the NJ method with 2000 bootstrap replicates. (**B**) Gene structures of *SlSut*, *DlSut*, and *NlSut*; orange boxes indicate CDS, cyan boxes the UTRs, and black lines the introns. (**C**) Conserved motifs of *SlSut*, *DlSut*, and *NlSut*; the numbers 1–20 and the differently colored boxes indicate the motifs. (**D**) Cis-acting elements in the promoter regions of *SUT*s. Statistical histogram of the number of biotic or abiotic stress response components, elements related to growth and development, light response elements, and response to hormone element.

**Figure 4 plants-13-00095-f004:**
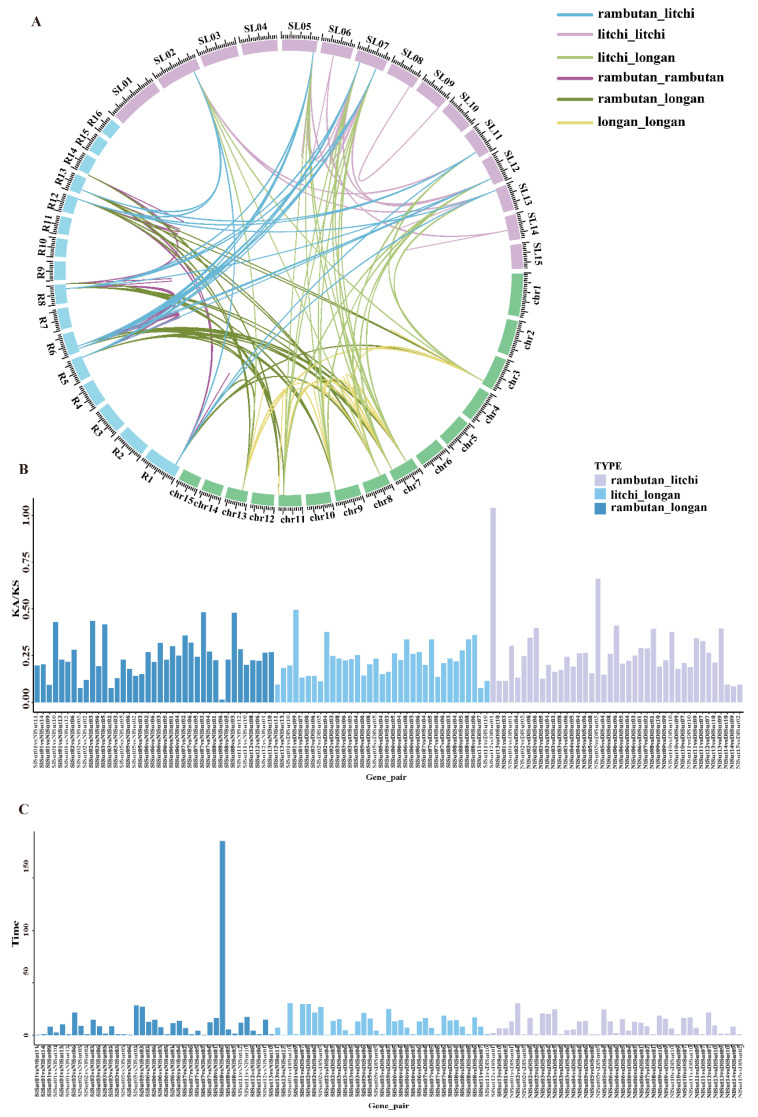
Collinearity analyses of the *SUT family genes*. (**A**) Circle plot of orthologous *SUT family gene* pairs among the three Sapindaceae species. The number on the scale represents the physical location of each chromosome. (**B**) Ks and Ka/Ks values of orthologous *SUT* family genes pairs between any two of the three Sapindaceae species. (**C**) Divergence time estimation of orthologous *SUT family gene* pairs between any two of the three Sapindaceae species.

**Figure 5 plants-13-00095-f005:**
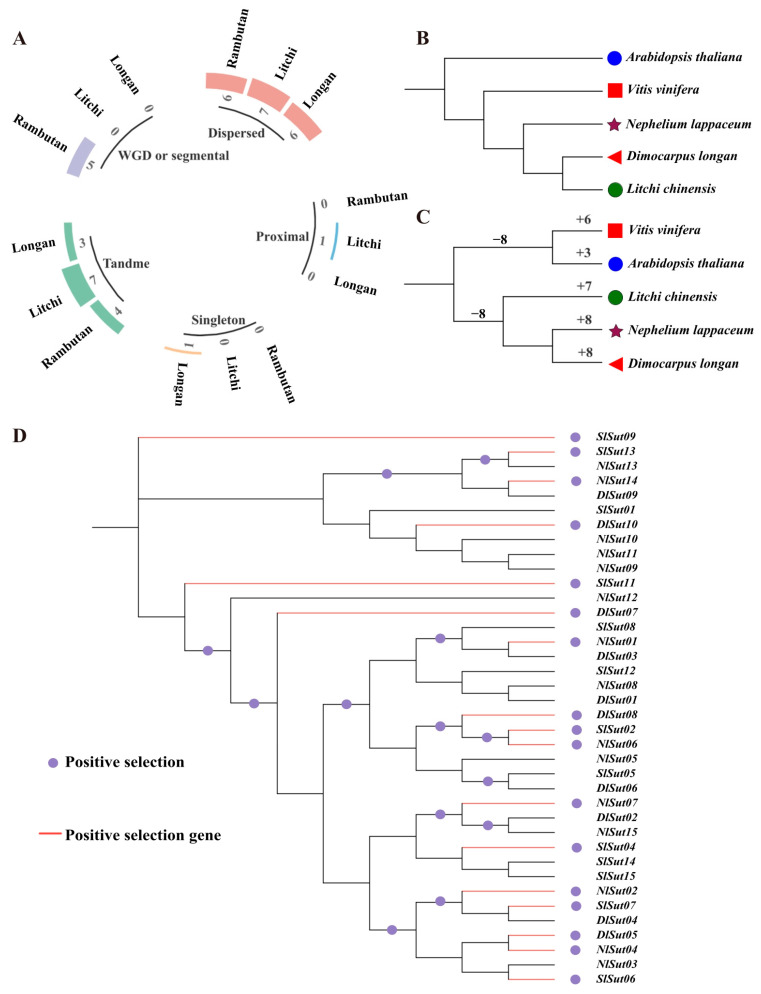
Duplication and positive selection analyses of the *SUT*s in the three *Sapindaceae* species. (**A**) Proportion of duplications in the whole genome and the *SUT*s. (**B**) Phylogenetic tree constructed using proteins encoded by the entire genome of the five species: litchi, longan, rambutan, grape, and *Arabidopsis.* (**C**) Phylogenetic tree constructed using proteins encoded by the *SUT*s of the five species: litchi, longan, rambutan, grape, and *Arabidopsis.* (**D**) Positive selection analysis of the *SUT*s, with the red stars representing the positively selected branches. The maximum likelihood (ML) phylogenetic tree was constructed employing PhyML (V3.1) software. The bootstrap values are indicated by the numbers on each branch.

**Figure 6 plants-13-00095-f006:**
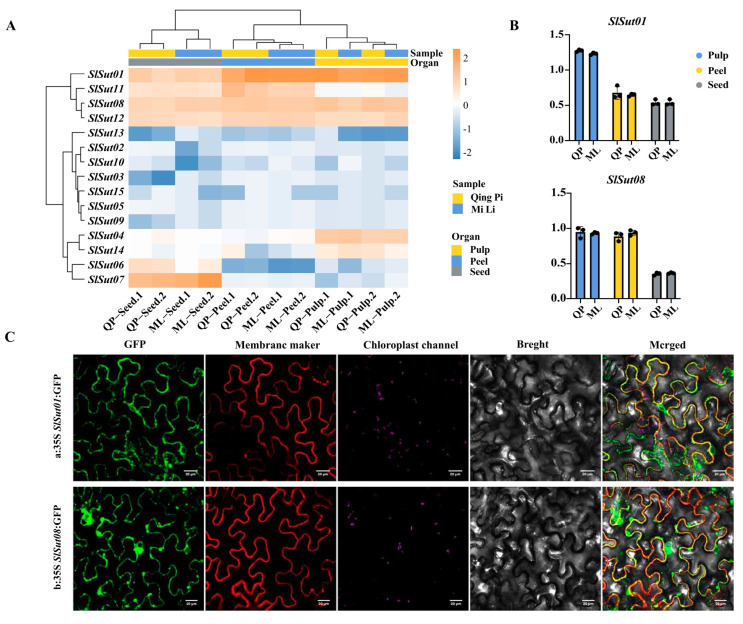
(**A**) Heatmap indicating the expression levels of *SUT*s in the pulp, peels, and seeds of QingPi (QP) and MiLi (ML). (**B**) qRT-PCR-based expression of *SlSut01* and *08* in these organs. (**C**) Fluorescent microscopic images of *SlSut01* and *08* in tobacco leaves. a, 35S:*SlSut01*:GFP; b, 35S:*SlSut08*:GFP.

## Data Availability

The RNA-seq raw data were submitted to the Sequence Read Archive (SRA) database at NCBI under BioProject number PRJNA989351 under the following link: https://www.ncbi.nlm.nih.gov/bioproject/PRJNA989351 (accessed on 14 August 2023).

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
