# Peer review of "Genome-Wide Identification and Expression Analysis of the SUT Family from Three Species of Sapindaceae Revealed Their Role in the Accumulation of Sugars in Fruits"

_plants, 2023, doi:10.3390/plants13010095_

Round 1

Reviewer 1 Report

Comments and Suggestions for Authors

The manuscript ‘Genome-wide identification and expression analysis of the  SUT family from three species of Sapindaceae revealed their role in the accumulation of sugars in fruits’ is a research paper dealing with the description of SUTs, their crucial functions, their evolution, and theoretical basis for sugar accumulation in the fruits of Sapindaceae. The subject is a worthy topic of investigation and appropriate to answer certain questions related to genetical/biochemical changes occurring in fruit crops.

The manuscript shows some little shortcomings. Please refer to them for improvements to the manuscript. Below I point out the main remarks.

1.      The part about subcellular localizations is not properly prepared and should be removed from the manuscript. The obtained results are at the tissue level not at the subcellular level. TEM analyses are needed. Authors should focus only on a genetic exploration of SUTs.

2.      Discussion: „proteins were localized and functioned in the cell membranÄ™”, „were also identified in other organelles, such as the endoplasmic reticulum or cytoplasm” Its overinterpretation. In this studies, authors do not observe it.

3.      Work organization should be improved, according to author's instructions. Figures wit results should be located near their description in the manuscript.

However, this approach is somehow original. Altogether it is well well-written and concise paper. In my opinion, the manuscript can be accepted for publication after minor revision.  

Author Response

Q1. The part about subcellular localizations is not properly prepared and should be removed from the manuscript. The obtained results are at the tissue level not at the subcellular level. TEM analyses are needed. Authors should focus only on a genetic exploration of SUTs.

Response: Thanks for your careful reading and constructive suggestions. TEM analysis is a valuable analysis in this article. After consulting, we found the experiment need fresh samples to operation. We all know litchi fruit mature period is every year on April to July. Therefore, it is not possible to obtain fresh litchi samples. We are sorry that we cannot provide the experimental results for the time being. But we will pay attention to this issue in the future work. And according to the Reviewer’s comments. We removed "But there is also a small amount of expression in other sites. This means that more and better research is needed." in result 2.6.

Response: We have made correction according to the Reviewer’s comments. We changed the last paragraph of the discussion part to "This study forms a basis for future comprehensive functional validation studies of these genes. The qRT-PCR and subcellular localization analysis indicated that two genes, SlSut01 and 08, were present on the cell membrane and played key roles in the accumulation of sugars in fruits."

Q2. Discussion: „proteins were localized and functioned in the cell membranÄ™”, „were also identified in other organelles, such as the endoplasmic reticulum or cytoplasm” Its overinterpretation. In this studies, authors do not observe it.

Q3. Work organization should be improved, according to author's instructions. Figures wit results should be located near their description in the manuscript.

Response: We have made correction according to the Reviewer’s comments. The figures of results are all located near their descriptions in the manuscript.

Reviewer 2 Report

Comments and Suggestions for Authors

In the manuscript by Jiang et al – (Genome-wide identification and expression analysis of the SUT family from three species of Sapindaceae revealed their role in the accumulation of sugars in fruits”, authors have applied bioinformatic and in vito tools to explain the role of sugar accumulation in Sapindaceae family fruits.

This manuscript has some major drawbacks, mentioned below, which must be corrected before it is accepted for publication.

Q1. Describe the terms: Vitis, AT, NISut, SiSut, DISut in legend of Fig1.

Q2. In Fig. 2, Description of C and D missing.

Q3. Fig 3 is missing.

Q4. In supplementary section S17, information is written in probably in Chinese language. Please change into in English.

Comments on the Quality of English Language

English language need to be improved.

Author Response

Q1. Describe the terms: Vitis, AT, NISut, SiSut, DISut in legend of Fig1.

Response: Thanks for your careful reading and constructive suggestions. We have rewritten legend of Fig1, and the revised content is as flowing: “Phylogenetic analysis of SUT proteins from five species: Litchi chinensis (SlSut), Dimocarpus longan (DlSut), Nephelium lappaceum (NlSut), Vitis vinifera (Vitis), and Arabidopsis thaliana (AT).

Q2. In Fig. 2, Description of C and D missing.

Response: Thanks for your comments to our work! We added "(C)Description of what is contained in the third panel" to the legend of Fig2. There is no D in fig2.

Q3. Fig 3 is missing.

Response: We have made correction according to the Reviewer’s comments. The missed of Figure 3 may be caused by formatting errors during file uploads.

Q4. In supplementary section S17, information is written in probably in Chinese language. Please change into in English.

Response: We have made correction according to the Reviewer’s comments. The information in supplementary section S17 were corrected as “Target gene”, “Internal reference gene”, “Primer”.

Finally, The English language was modified for the full manuscript.

Round 2

Reviewer 2 Report

Comments and Suggestions for Authors

Now, the manuscript is recommended to publish.